# Computationally Efficient Soft Detection Schemes for Coded Massive MIMO Systems †

**Meixiang Zhang [1], Zhi Zhang [1], Satya Chan [2]** and **Sooyoung Kim [2],***

[1]   School of Information Engineeing, Yangzhou University, Yangzhou 225009, China;
      mxzhang@yzu.edu.cn (M.Z.); MX120170410@yzu.edu.cn (Z.Z.)
[2]   Division of Electronic Engineering, IT Convergence Research Center, Jeonbuk National University,
      Jeonju 54896, Korea; csatya@jbnu.ac.kr
*    Correspondence: sookim@jbnu.ac.kr; Tel.: +82-63-270-3992
†    This paper is an extended version of our paper published in ICTC 2019.

**Abstract:**    This paper presents a computationally efficient soft detection scheme for massive multiple-input multiple-output (MIMO) systems. The proposed scheme adopts joint iterative detection and decoding (JIDD) methods for their capacity limiting performances. In addition, the minimum mean square error parallel interference cancellation (MMSE-PIC)-based detection scheme is used for soft information exchange. We propose a number of techniques to reduce the computational complexity, while keeping almost the same performance as the conventional ones. First, a technique is proposed to approximate the Gram matrix to a constant valued diagonal matrix. This proposal can lead to elimination of complex matrix inversion process and multiple layer dependent estimations, resulting in huge complexity reduction. Second, compact equations to estimate soft-symbol values for $M$-ary (quadrature amplitude modulation) QAM are derived. From the investigation example of $2^8$-QAM in this paper, this proposal showed more than two orders of less computations compared to the conventional scheme. The simulation results demonstrate that the proposed method can achieve approximating performance to the conventional method with a largely reduced computational complexity.

**Keywords:** massive MIMO; soft detection; JIDD; coded MIMO; MMSE-PIC

## 1. Introduction

In recent years, the massive multiple-input multiple-output (MIMO) technology which utilizes hundreds of antennas at the base station (BS) has attracted great interests due to its high data rates for a given bandwidth [1]. In coded MIMO systems, the joint iterative detection and decoding (JIDD) method can produce high performance gain [2]. The performance gain from the JIDD method could be achieved by iteratively exchanging soft information between the maximum likelihood (ML) detector and decoder, at the cost of computational complexity. Minimum mean square error (MMSE)-based detection schemes were often considered for JIDD due to their reasonable complexity and performance tradeoff. These were called the minimum mean square error parallel interference cancellation (MMSE-PIC) methods [3].

A complexity reduced JIDD scheme with the MMSE-PIC was proposed in [4], and later JIDD with three loops were proposed in [5]. Even though the MMSE-PIC method for JIDD provides a trade-off between the performance and computational complexity, its computational complexity is still too high for massive MIMO systems. The main computational burden during the MMSE-PIC process is incurred from the estimation of the Gram matrix and matrix inversion [6]. Several attempts were made to apply the MMSE-PIC based detection for massive MIMO systems. Large efforts were made to

approximate matrix inversion process by using iteration based methods [7–10]. In addition, Neumann series expansion (NSE) method [11] and Newton iteration method [12] were proposed.

Considering that the channel matrix is asymptotically orthogonal in massive MIMO systems [13], a diagonal-like matrix was proposed to approximate the MMSE filtering matrix to reduce the complexity [14]. However, these methods still suffer from too much complexity when the number of antennas at the BS is massive. Not only the the complexity caused by calculating the Gram matrix and the matrix inversion, but also the post-equalization signal-to-interference-plus-noise ratio (PE-SINR) estimation at every layer causes a lot of computations [10]. For this reason, our previous study presented a complexity reduced method to approximate PE-SINR, but the investigation was limited to non-iterative detection and decoding schemes [15].

In this paper, we propose an efficient MMSE-PIC-based JIDD scheme with huge reduction in computational complexity for coded massive MIMO systems. The proposed scheme first derives a universal constant diagonal matrix which eventually results in a simplified filtering matrix as well as compact PE-SINR estimation. In addition, an efficient soft-symbol estimation method is proposed to reduce the number of computations. Especially, we present compact equations for *M*-ary quadrature amplitude modulation (QAM) schemes which require less than 1% of the computational complexity compared to the conventional scheme. Finally, soft bit information (SBI) to the decoder is extracted by using a simple symbol mapping method to further reduce the complexity.

The remainder of this paper is organized as follows. In Section 2, we first review an existing massive MIMO system with JIDD combined with MMSE-PIC, and introduce a simple soft demapping method which can extract soft bit information with a linear-order complexity. Section 3 presents the proposed MMSE-PIC-based JIDD scheme with a number of techniques to reduce the computational complexity. In addition, the complexity reduced equations to estimate soft symbols for MMSE-PIC process are derived. Simulation results are shown in Section 4, and Section 5 concludes the paper.

*Notation*: Lowercase and uppercase boldface letters denote vectors and matrices, respectively, lower and upper case letters denote scalars. Transpose, conjugate transpose, matrix inversion, and norm operations are denoted by $(\cdot)^T$, $(\cdot)^H$, $(\cdot)^{-1}$ and $|\cdot|$, respectively. In addition, $M \times M$ identity matrix is represented by $\mathbf{I}_M$.

## 2. Related Works

### 2.1. Uplink System Model with JIDD

In this paper, we consider a coded massive MIMO system equipped with $N$ receive antennas at the BS for $M$ transmit antennas ($N >> M$), as shown in Figure 1. At the transmitter, we encode the information vector $\mathbf{u}$ to produce the codeword $\mathbf{c}$. Then, $M \times K$ codewords are interleaved by an interleaver and are modulated successively before going through the channel, where $K$ denotes the number of bits per transmitted symbol. Every $MK$ bits from the interleaver, i.e., $\mathbf{x} = [x_{1,1}, \cdots, x_{1,K}, x_{2,1}, \cdots, x_{m,k}, \cdots, x_{M,K}]$ are grouped and modulated to the transmitted symbol vector $\mathbf{s} = [s_1, s_2, \cdots, s_m, \cdots, s_M]^T$, where $x_{m,k}$ is the $k$th bit of the transmitted symbol from the $m$th antenna, $s_m$, which is independently mapped from a complex constellation $O$. The received symbol vector $\mathbf{y} = [y_1, y_2, \cdots, y_m, \cdots, y_N]^T$ can be represented as follows:

$$\mathbf{y} = \mathbf{Hs} + \mathbf{n}, \tag{1}$$

where $\mathbf{H}$ is an $N \times M$ complex-valued channel matrix whose entries are independent and identically distributed with zero mean and unit variance, and $\mathbf{n}$ is Gaussian noise vector whose elements are independent zero-mean complex Gaussian random variables with variance $\sigma^2$ per dimension.

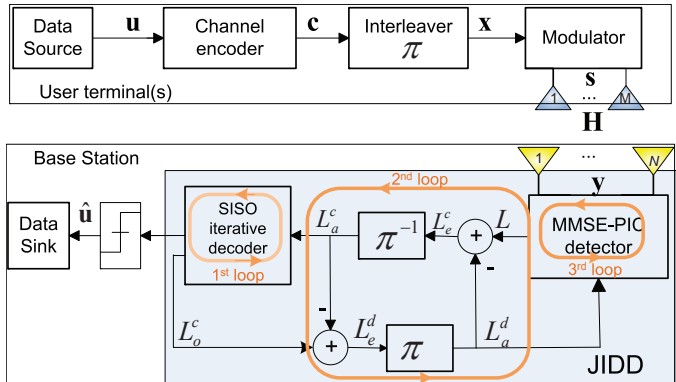

**Figure 1.** Coded multiple-input multiple-output (MIMO) system with minimum mean square error parallel interference cancellation (MMSE-PIC)-based joint iterative detection and decoding (JIDD).

At the receiver with JIDD scheme, there are three loops working together to improve the overall performance. Loop 1 denotes the iterations inside the channel decoder, Loop 2 is used to exchange soft information between the detector and decoder, and Loop 3 is inside the MMSE-PIC detector. First, SBI $L$ is calculated by the MMSE-PIC detector. A detailed process to estimate $L$ can be found in [4], and it is briefly discussed in Section 2.2. Afterwards, $L$ is subtracted by the *a priori* information, $L_a^d$ from the channel decoder. Then, the extrinsic information is produced; $L_e^c = L - L_a^d$. $L_e^c$ is passed through the de-interleaver and its de-interleaved version, $L_a^c$ is used by the channel decoder as the *a priori* information. After decoding, the soft output $L_o^c$ is generated and the extrinsic information for MMSE-PIC is calculated by $L_e^d = L_o^c - L_a^c$. In addition, its interleaved version, $L_a^d$, is fed back to the MMSE-PIC detector as the *a priori* information.

### 2.2. MMSE-PIC Detection for MIMO System

The MMSE-PIC detector in Figure 1 needs to estimate soft symbol values in order to incorporate soft information from the channel decoder. The first step is to estimate the expected soft-symbol value $\widetilde{s}_i$ and the variance $E_i$ of the transmitted symbol $s_i$, by using the *a priori* information, as follows [3,14]:

$$\widetilde{s}_i = \sum_{a \in O} \frac{a}{2^K} \prod_{k=1}^{K} (1 + \widetilde{x}_{i,k} \zeta_{i,k}), \tag{2}$$

$$E_i = \sum_{a \in O} \frac{|a|^2}{2^K} \prod_{k=1}^{K} (1 + \widetilde{x}_{i,k} \zeta_{i,k}) - |\widetilde{s}_i|^2, \tag{3}$$

where $a$ is a constellation symbol of $O$, $\widetilde{x}_{i,k}$ is set to be $-1$ and 1 according to the $k$th bits of $a$, and $\zeta_{i,k}$ is considered as zero at the first stage, and in the following iterations, $\zeta_{i,k}$ can be calculated as:

$$\zeta_{i,k} = \tanh((L(x_{i,k}) + L_a^d(x_{i,k}))/2). \tag{4}$$

Then, we perform the parallel interference cancellation (PIC) process on the received symbol vector to cancel the interference between layers, and the interference-cancelled symbol vector for the $i$th layer, $\hat{\mathbf{y}}_i$ is calculated by using the following equation:

$$\hat{\mathbf{y}}_i = \mathbf{y} - \sum_{j \neq i} \mathbf{h}_j \widetilde{s}_j = \mathbf{h}_i s_i + \widetilde{\mathbf{n}}, \tag{5}$$

where $\mathbf{h}_i$ denotes the $i$th column of $\mathbf{H}$, and $\widetilde{\mathbf{n}} = \sum_{j \neq i} \mathbf{h}_j (s_i - \widetilde{s}_i) + \mathbf{n}$.

The next step is the suppression of noise plus interference (NPI) term in (5) with the following equation:

$$\hat{z}_i = \mathbf{w}_i^H \hat{\mathbf{y}}_i, \tag{6}$$

where $\mathbf{w}_i^H$ is the $i$th row of the MMSE filtering matrix, $\mathbf{W}^H$, which is calculated as follows [16]:

$$\mathbf{W}^H = (\mathbf{H}^H \mathbf{H} \mathbf{\Lambda} + \sigma^2 \mathbf{I}_M)^{-1} \mathbf{H}^H = \widetilde{\mathbf{W}}^{-1} \mathbf{H}^H, \tag{7}$$

where $\mathbf{\Lambda}$ is a diagonal matrix whose diagonal element $\lambda_{i,i}$ equals to $E_i$ calculated in (3).

For the estimation of SBI in the form of log-likelihood ratio (LLR) to the decoder, the channel-compensated value, $z_i = \hat{z}_i / (\mathbf{w}_i^H \mathbf{h}_i)$ needs to be used as follows [16], so that the SBI estimation is not subject to the channel gain.

$$L(x_{i,k}|\mathbf{y}, \mathbf{H}) \approx \min_{a \in O_k^0} \left( \rho_i |z_i - a|^2 + \sum_{k=1}^{K} \ln \left( 1 + e^{(-x_{i,k} L_a^d(x_{i,k}))} \right) \right)$$
$$- \min_{a \in O_k^1} \left( \rho_i |z_i - a|^2 + \sum_{k=1}^{K} \ln \left( 1 + e^{(-x_{i,k} L_a^d(x_{i,k}))} \right) \right), \tag{8}$$

where $a \in O_k^0$ and $a \in O_k^1$ represent the constellation symbols with the $k$th bit of 0 and 1, respectively, and the PE-SINR for the $i$th transmitted symbol can be estimated as:

$$\rho_i = \frac{\mu_i}{1 - E_i \mu_i}, \tag{9}$$

where

$$\mu_i = \mathbf{w}_i^H \mathbf{h}_i. \tag{10}$$

Estimation of SBI using (8) requires exhaustive search to find the minima. A complexity reduced scheme which eliminates the search process to find the minima can be used as follows [17]:

$$L(x_{i,k}|\mathbf{y}, \mathbf{H}) \approx \omega_{i,k} \rho_i \left( \left| \oslash(z_i, \epsilon_k^0) - q_k^0 \right|^2 - \left| \oslash(z_i, \epsilon_k^1) - q_k^1 \right|^2 \right), \tag{11}$$

where $\omega_{i,k}$ is a sign function introduced by symbol mapping, and $q_k^b$ is the unique symbol nearest to the mapped $z_i$. In addition, $\oslash(z_i, \epsilon_k^b) = |z_i| e^{j\epsilon_k^b}$, where $\epsilon_k^b$, $b \in \{0,1\}$ represents the phase of mapped symbol for the $k$th bit of $b$.

## 3. Proposed Method

In this section, we propose to approximate the Gram matrix with a derived constant diagonal matrix to reduce the computational complexity of the MMSE-PIC-based JIDD scheme. With the proposed scheme, the computations of estimating the Gram-matrix, filtering matrix and PE-SINR are highly reduced. In addition, we derive compact equations to estimate soft-symbol values, and then expand the results for well-known $M$-ary QAM with highly reduced complexity, compared to (2) and (3). Finally, the symbol mapping technique in (11) is tailored to the proposed scheme for further complexity reduction.

### 3.1. MMSE-Filtering-Matrix Approximation

In this subsection, we focus on reducing the computational complexity of the MMSE-PIC-based JIDD scheme caused by the calculation of the Gram matrix, filtering matrix, and PE-SINR. First, by using the property that the Gram matrix, $\mathbf{G} = \mathbf{H}^H \mathbf{H}$, can be approximated to a diagonal-like matrix in a massive MIMO system, $\mathbf{W}^H$ can also be approximated as follow [13,14]:

$$\mathbf{W}^H = (\mathbf{G}\boldsymbol{\Lambda} + \sigma^2 \mathbf{I}_M)^{-1} \mathbf{H}^H$$

$$\approx \begin{bmatrix} \widetilde{w}_{1,1} & 0 & \cdots & 0 \\ 0 & \widetilde{w}_{2,2} & \cdots & 0 \\ 0 & 0 & \ddots & 0 \\ 0 & 0 & 0 & \widetilde{w}_{M,M} \end{bmatrix}^{-1} \mathbf{H}^H, \tag{12}$$

where $\widetilde{w}_{i,i} = g_{i,i} E_i + \sigma^2$, and $g_{i,i}$ denotes the $i$th diagonal element of $\mathbf{G}$. This method reduces the complexity of calculating matrix inversion and produces an approximating performance to the conventional MMSE-PIC-based JIDD system. Nevertheless, this method still requires not a small amount of computational complexity to calculate diagonal elements of $\mathbf{G}$.

A previous study proposed a method to reduce the computational complexity of the PE-SINR estimation, by approximating the diagonal elements of $\mathbf{G}$ with a universal constant value [15]. However, this method cannot be directly applied to the JIDD system. To solve this problem, we propose an approximating method in the calculation of both the Gram matrix and PE-SINR as follows. We use the fact that the column vectors of the channel matrix are almost orthogonal as the number of antennas increases [13]. Then $\mathbf{G}$ can be approximated by $N/M\mathbf{I}_M$, by considering transmit power normalization. Consequently, the filtering matrix $\mathbf{W}^H$ can be estimated without any matrix inversion as follows:

$$\mathbf{W}^H = (\mathbf{G}\boldsymbol{\Lambda} + \sigma^2 \mathbf{I}_M)^{-1} \mathbf{H}^H$$

$$\approx \begin{bmatrix} \frac{N}{M}E_1 + \sigma^2 & 0 & \cdots & 0 \\ 0 & \frac{N}{M}E_2 + \sigma^2 & \cdots & 0 \\ 0 & 0 & \ddots & 0 \\ 0 & 0 & 0 & \frac{N}{M}E_M + \sigma^2 \end{bmatrix}^{-1} \mathbf{H}^H. \tag{13}$$

$$= \begin{bmatrix} (\frac{N}{M}E_1 + \sigma^2)^{-1} & 0 & \cdots & 0 \\ 0 & (\frac{N}{M}E_2 + \sigma^2)^{-1} & \cdots & 0 \\ 0 & 0 & \ddots & 0 \\ 0 & 0 & 0 & (\frac{N}{M}E_M + \sigma^2)^{-1} \end{bmatrix} \mathbf{H}^H.$$

With the above approximation, $\mathbf{w}_i^H$ in (6) can be expressed as follows:

$$\mathbf{w}_i^H \approx \frac{1}{\frac{N}{M}E_i + \sigma^2} \mathbf{h}_i^H, \tag{14}$$

where, $\mathbf{h}_i^H$ is the $i$th row of $\mathbf{H}^H$. Therefore, (6) can be simplified as follows:

$$\hat{z}_i = \mathbf{w}_i^H \hat{\mathbf{y}}_i \approx \frac{1}{\frac{N}{M}E_i + \sigma^2} \mathbf{h}_i^H \hat{\mathbf{y}}_i, \tag{15}$$

Finally, we show that the PE-SINR estimation can be simplified to a layer independent universal value. First, $\mu_i$ in (10) can be expressed as follows, by inserting (14).

$$\mu_i = \mathbf{w}_i^H \mathbf{h}_i \approx \frac{1}{\frac{N}{M}E_i + \sigma^2} \mathbf{h}_i^H \mathbf{h}_i = \frac{\frac{N}{M}}{\frac{N}{M}E_i + \sigma^2}. \tag{16}$$

Then, the layer dependent PE-SINR in (9) can be expressed as a constant value as follows:

$$\rho_i \approx \frac{N}{M\sigma^2}. \tag{17}$$

### 3.2. Soft-Symbol Value Estimation

Direct estimation of (2) and (3) requires exponentially increasing number of computations by the number of bits in a symbol. If we use symmetric property of QAM constellations, then we can highly reduce the complexity. For this, we first decompose the complex value $\widetilde{s}_i$ into real and imaginary parts as follows:

$$\widetilde{s}_i = \Re(\widetilde{s}_i) + j\Im(\widetilde{s}_i), \tag{18}$$

where $\Re(x)$ and $\Im(x)$ are the real and imaginary parts of the complex number $x$.

With the above decomposition, computations in (2) is now performed with real number operations, and $\Re(\widetilde{s}_i)$ can be derived as follows.

$$
\begin{aligned}
\Re(\widetilde{s}_i) &= \sum_{a \in O} \frac{\Re(a)}{2^K} \prod_{k=1}^{K} (1 + \widetilde{x}_{i,k}\zeta_{i,k}) \\
&= \sum_{\Re(a) \in O^R} \sum_{\Im(a) \in O^I} \frac{\Re(a)}{2^K} \prod_{k \in \kappa^R} (1 + \widetilde{x}_{i,k}\zeta_{i,k}) \prod_{k \in \kappa^I} (1 + \widetilde{x}_{i,k}\zeta_{i,k}) \\
&= \sum_{\Re(a) \in O^R} \frac{\Re(a)}{2^K} \prod_{k \in \kappa^R} (1 + \widetilde{x}_{i,k}\zeta_{i,k}) \sum_{\Im(a) \in O^I} \prod_{k \in \kappa^I} (1 + \widetilde{x}_{i,k}\zeta_{i,k}) \\
&= \sum_{\Re(a) \in O^R} \frac{\Re(a)}{2^K} \prod_{k \in \kappa^R} (1 + \widetilde{x}_{i,k}\zeta_{i,k}) 2^{K/2} \\
&= \sum_{\Re(a) \in O^R} \frac{\Re(a)}{2^{K/2}} \prod_{k \in \kappa^R} (1 + \widetilde{x}_{i,k}\zeta_{i,k}),
\end{aligned}
\tag{19}
$$

where $O^R = \{a|a \text{ is a symbol projected on to the real axis}\}$, $\kappa^R$ represents a set of bit indexes of the symbols which are projected to real axis. As shown in (19), the number of computations in $\prod_{k=1}^{K}(1 + \widetilde{x}_{i,k}\zeta_{i,k})$ of (2) is reduced from $2^K$ to $2^{K/2}$.

We derive $\Im(\widetilde{s}_i)$ with the same way as follows:

$$
\begin{aligned}
\Im(\widetilde{s}_i) &= \sum_{a \in O} \frac{\Im(a)}{2^K} \prod_{k=1}^{K} (1 + \widetilde{x}_{i,k}\zeta_{i,k}) \\
&= \sum_{\Im(a) \in O^I} \frac{\Im(a)}{2^{K/2}} \prod_{k \in \kappa^I} (1 + \widetilde{x}_{i,k}\zeta_{i,k}),
\end{aligned}
\tag{20}
$$

where $O^I = \{a|a \text{ is a symbol projected on to the imaginary axis}\}$, $\kappa^I$ represent the bit indexes of the symbols which are projected to imaginary axis.

Now we prove that $E_i$ can be decomposed into $E_i^R$ and $E_i^I$ which are the variances estimated for $\Re(s_i)$ and $\Im(s_i)$, respectively, as follows:

$$
\begin{aligned}
E_i &= \sum_{a \in O} \frac{|a|^2}{2^K} \prod_{k=1}^{K} (1 + \widetilde{x}_{i,k}\zeta_{i,k}) - |\widetilde{s}_i|^2 \\
&= \sum_{\Re(a) \in O^R} \sum_{\Im(a) \in O^I} \frac{\Re(a)^2 + \Im(a)^2}{2^K} \prod_{k \in \kappa^R} (1 + \widetilde{x}_{i,k}\zeta_{i,k}) \prod_{k \in \kappa^I} (1 + \widetilde{x}_{i,k}\zeta_{i,k}) - |\widetilde{s}_i|^2 \\
&= \sum_{\Re(a) \in O^R} \sum_{\Im(a) \in O^I} \frac{\Re(a)^2}{2^K} \prod_{k \in \kappa^R} (1 + \widetilde{x}_{i,k}\zeta_{i,k}) \prod_{k \in \kappa^I} (1 + \widetilde{x}_{i,k}\zeta_{i,k}) \\
&\quad + \sum_{\Re(a) \in O^R} \sum_{\Im(a) \in O^I} \frac{\Im(a)^2}{2^K} \prod_{k \in \kappa^R} (1 + \widetilde{x}_{i,k}\zeta_{i,k}) \prod_{k \in \kappa^I} (1 + \widetilde{x}_{i,k}\zeta_{i,k}) - |\widetilde{s}_i|^2 \\
&= \sum_{\Re(a) \in O^R} \frac{\Re(a)^2}{2^{K/2}} \prod_{k \in \kappa^R} (1 + \widetilde{x}_{i,k}\zeta_{i,k}) - \Re(\widetilde{s}_i)^2 + \sum_{\Im(a) \in O^I} \frac{\Im(a)^2}{2^{K/2}} \prod_{k \in \kappa^I} (1 + \widetilde{x}_{i,k}\zeta_{i,k}) - \Im(\widetilde{s}_i)^2 \\
&= E_i^R + E_i^I,
\end{aligned}
\tag{21}
$$

where

$$E_i^R = \sum_{\Re(a)\in O^R} \frac{\Re(a)^2}{2^{K/2}} \prod_{k\in\kappa^R}(1+\widetilde{x}_{i,k}\zeta_{i,k}) - \Re(\widetilde{s}_i)^2,$$

$$E_i^I = \sum_{\Im(a)\in O^I} \frac{\Im(a)^2}{2^{K/2}} \prod_{k\in\kappa^I}(1+\widetilde{x}_{i,k}\zeta_{i,k}) - \Im(\widetilde{s}_i)^2. \tag{22}$$

With the above decomposition, we further reduce the computational complexity of (22) by expanding $E_i^R$ as follows:

$$
\begin{aligned}
E_i^R &= \sum_{\Re(a)\in O^R} \frac{\Re(a)^2}{2^{K/2}} \prod_{k\in\kappa^R}(1+\widetilde{x}_{i,k}\zeta_{i,k}) - \Re(\widetilde{s}_i)^2 \\
&= \sum_{\Re(a)\in O^{R+}} \frac{\Re(a)^2}{2^{K/2}}(1+\zeta_{i,\chi}) \prod_{k\in\kappa^{R+}}(1+\widetilde{x}_{i,k}\zeta_{i,k}) \\
&\quad + \sum_{\Re(a)\in O^{R-}} \frac{\Re(a)^2}{2^{K/2}}(1-\zeta_{i,\chi}) \prod_{k\in\kappa^{R-}}(1+\widetilde{x}_{i,k}\zeta_{i,k}) - \Re(\widetilde{s}_i)^2 \\
&= \sum_{\Re(a)\in O^{R+}} \frac{\Re(a)^2}{2^{K/2}} 2 \prod_{k\in\kappa^{R+}}(1+\widetilde{x}_{i,k}\zeta_{i,k}) - \Re(\widetilde{s}_i)^2 \\
&= \sum_{\Re(a)\in O^{R+}} \frac{\Re(a)^2}{2^{K/2-1}} \prod_{k\in\kappa^{R+}}(1+\widetilde{x}_{i,k}\zeta_{i,k}) - \Re(\widetilde{s}_i)^2,
\end{aligned}
\tag{23}
$$

where $O^{R+} = \{a|a$ is a symbol projected on to the real axis, $\Re(a) > 0\}$, $O^{R-} = \{a|a$ is a symbol projected on to the real axis, $\Re(a) < 0\}$, $\kappa^{R+} = \kappa^{R-} = \{k|k \in \kappa^R, k \neq \chi\}$, and $\chi$ is the bit index that determines whether $a$ is included in $O^{R+}$ or $O^{R-}$. Likewise,

$$
\begin{aligned}
E_i^I &= \sum_{\Im(a)\in O^I} \frac{\Im(a)^2}{2^{K/2}} \prod_{k\in\kappa^I}(1+\widetilde{x}_{i,k}\zeta_{i,k}) - \Im(\widetilde{s}_i)^2 \\
&= \sum_{\Im(a)\in O^{I+}} \frac{\Im(a)^2}{2^{K/2-1}} \prod_{k\in\kappa^{I+}}(1+\widetilde{x}_{i,k}\zeta_{i,k}) - \Im(\widetilde{s}_i)^2,
\end{aligned}
\tag{24}
$$

where $O^{I+} = \{a|a$ is a symbol projected on to the imaginary axis, $\Im(a) > 0\}$, and $\kappa^{I+} = \{k|k \in \kappa^I, k \neq \chi\}$. By expanding (19)–(24) for a specific Gray coded QAM constellation, we could further reduce the number of computations. Table 1 shows the results of the expansions using (19)–(24) for a Gray coded 16, 64, and 256-QAM constellations in [17].

**Table 1.** Complexity reduced estimations of $\widetilde{s}_i$ and $E_i$ by expanding (19)–(24) for a Gray coded quadrature amplitude modulation (QAM).

| $K$ | $A$ | $\widetilde{s}_i$ | $E_i$ |
|---|---|---|---|
| 4 | $1/\sqrt{10}$ | $A(-2\zeta_{i,1} + \zeta_{i,1}\zeta_{i,3}) + jA(-2\zeta_{i,2} + \zeta_{i,2}\zeta_{i,4})$ | $A^2(10 - 4\zeta_{i,3} - 4\zeta_{i,4}) - |\widetilde{s}_i|^2$ |
| 6 | $1/\sqrt{42}$ | $A(-4\zeta_{i,1} + 2\zeta_{i,1}\zeta_{i,3} - \zeta_{i,1}\zeta_{i,3}\zeta_{i,5})$ $+jA(-4\zeta_{i,2} + 2\zeta_{i,2}\zeta_{i,4} - \zeta_{i,2}\zeta_{i,4}\zeta_{i,6})$ | $A^2(42 - 16\zeta_{i,3} - 4\zeta_{i,5} + 8\zeta_{i,3}\zeta_{i,5}$ $-16\zeta_{i,4} - 4\zeta_{i,6} + 8\zeta_{i,4}\zeta_{i,6}) - |\widetilde{s}_i|^2$ |
| 8 | $1/\sqrt{170}$ | $A(-8\zeta_{i,1} + 4\zeta_{i,1}\zeta_{i,3} - 2\zeta_{i,1}\zeta_{i,3}\zeta_{i,5} + \zeta_{i,1}\zeta_{i,3}\zeta_{i,5}\zeta_{i,7})$ $+jA(-8\zeta_{i,2} + 4\zeta_{i,2}\zeta_{i,4} - 2\zeta_{i,2}\zeta_{i,4}\zeta_{i,6} + \zeta_{i,2}\zeta_{i,4}\zeta_{i,6}\zeta_{i,8})$ | $A^2(170 - 64\zeta_{i,3} - 16\zeta_{i,5} - 4\zeta_{i,7}$ $+32\zeta_{i,3}\zeta_{i,5} + 8\zeta_{i,5}\zeta_{i,7} - 16\zeta_{i,3}\zeta_{i,5}\zeta_{i,7}$ $-64\zeta_{i,4} - 16\zeta_{i,6} - 4\zeta_{i,8} + 32\zeta_{i,4}\zeta_{i,6}$ $+8\zeta_{i,6}\zeta_{i,8} - 16\zeta_{i,4}\zeta_{i,6}\zeta_{i,8}) - |\widetilde{s}_i|^2$ |

*3.3. Algorithm for the Proposed Methods and Complexity Comparisons*

With the proposed techniques, diagonal elements of the Gram matrix are approximately represented by a universal constant value, and the estimation of soft-symbol values are performed with

much lower complexity. Eventually, it results in that the computational complexity of matrix inversion is greatly reduced to a linear order, and the estimation of the PE-SINR is also greatly simplified. The complete process of the proposed method is summarized in Algorithm 1, where $\eta$ and $l$ denote the number of iterations between the MMSE-PIC detector and decoder, and the number of iterations inside the detector, respectively.

---

**Algorithm 1** Pseudo-code for the proposed complexity reduced MMSE-PIC-based JIDD algorithm

---

Initialize: $\mathbf{L}_a^d = 0$, $\mathbf{L} = 0$
**for** $1 \leq \eta \leq \eta_{max}$ **do**
　　**for** $1 \leq l \leq l_{max}$ **do**
　　　　Calculate $\zeta_{i,k}$ with (4), $\widetilde{s}_i$ and $E_i$ with Table 1
　　　　$\widetilde{w}_{i,i} = \frac{N}{M}E_i + \sigma^2$
　　　　$\hat{z}_i = \mathbf{h}_i^H \hat{\mathbf{y}}_i / \widetilde{w}_{i,i}$
　　　　$\mu_i = \frac{\frac{N}{M}}{\frac{N}{M}E_i + \sigma^2}$, and $\rho_i = \frac{N}{M\sigma^2}$
　　　　$L(x_{i,k}|\mathbf{y}, \mathbf{H})$ estimation with (11)
　　**end for**
　　Iterative decoding, output $\mathbf{L}_a^d$
**end for**

---

The computational complexity of the proposed method is compared with two conventional schemes in terms of the number of multiplications per iteration, and the results are shown in Table 2. We compare the computational complexity of major computations during the MMSE-PIC process, which are matrix-inversion processes as well as calculating $\mathbf{G}$, $\mu_i$, $\widetilde{s}_i$, and $E_i$ as listed in Table 2. As shown in the table, the computational complexity of matrix-inversion is reduced to a linear order in the proposed scheme, benefiting from the diagonal approximation. The complexity of the Gram-matrix estimation in the proposed scheme is reduced to $O(1)$ due to the further approximation with a universal constant value. Given $\mathbf{W}^{-1}$ and $\mathbf{G}$, the computational complexities of estimating $\mu_i$ in the conventional schemes are $O(M^2)$, while that of the proposed scheme is reduced to $O(M)$.

**Table 2.** Computational complexity comparisons.

| | Conventional [16] | Conventional [4] | Proposed |
|---|---|---|---|
| $()^{-1}$ | $O(M^3)$ | $O(M)$ | $O(M)$ |
| $\mathbf{G}$ | $O(M^2N)$ | $O(MN)$ | $O(1)$ |
| $\mu_i, \rho_i$ | $O(M^2),O(M)$ | $O(M),O(M)$ | $O(M),O(1)$ |
| $\widetilde{s}_i$ | $O(K2^K)$ | $O(K2^{K/2})$ | $O(K2^{K/2})$ |
| $E_i$ | $O(K2^K)$ | $O(K2^{K/2})$ | $O(K2^{K/2})$ |

Especially, we compare the complexity of calculating $\widetilde{s}_i$ and $E_i$ in more details. Table 3 compares the number of multiplications required in the conventional schemes using (2) and (3), with the proposed compact methods using (19)–(24). To estimate $\widetilde{s}_i$ with (2), we need $K - 1$ multiplications in $\prod_{k=1}^K$ for each constellation symbol $a$, in addition to three multiplications with $\Re(a)$, $\Im(a)$, and $1/2^K$, respectively, and thus in total we need $(K - 1 + 3)2^K$ multiplications. Similarly, to estimate $E_i$ in (3), we need $K - 1$ multiplications in $\prod_{k=1}^K$ for each constellation symbol $a$, in addition to two multiplications with $1/2^K$ and $|a|^2$, respectively, and then two multiplication for $|\widetilde{s}_i|^2$. Therefore, we need $((K - 1 + 2)2^K + 2)$ multiplications in total. On the other hand, the proposed method reduces the number of multiplications from $O(K)$ to $O(K/2)$, due to symbol projections to real and imaginary axes, respectively. Therefore, we need $2(K/2 - 1 + 2)2^{K/2}$ multiplications to estimate $\widetilde{s}_i$, and $(2(K/2 - 2 + 2)2^{K/2-1} + 2)$ multiplications to estimate $E_i$. We note that if we use the expanded results in Table 1, then the complexity is less than 1% of the direct estimation of (2) and (3).

**Table 3.** Complexity of estimating $\widetilde{s}_i$ and $E_i$ in terms of the number of multiplications.

| $K$ | (2), (3) | (19)+(20), (23)+(24) | $\widetilde{s}_i$, $E_i$ in Table 1 |
|---|---|---|---|
| 4 | 96, 82 | 24, 10 | 6, 5 |
| 6 | 512, 450 | 64, 26 | 12, 11 |
| 8 | 2560, 2306 | 160, 66 | 20, 23 |

## 4. Simulation Results

We simulated the performance of the coded MIMO system with 16 quadrature amplitude modulation (QAM) over a frequency-flat Rayleigh fading channel. As a forward error correction (FEC) scheme, low density parity check (LDPC) code with a length of 16,200 bits and a code rate of 1/3 was used. We first compare the bit error rate (BER) performance of various coded MIMO systems at the first iteration of MMSE-PIC detection. This is to see the performance behavior by the number of iterations inside of the LDPC decoder, before we investigate the performances of JIDD based detection schemes.

Figure 2 compares the BER performance of the proposed method with that of the conventional scheme for $M \times N$ MIMO systems. The maximum number of iterations inside the LDPC decoder is denoted by $\alpha$. As shown in the figure, the performance of both methods improves as $\alpha$ increases, regardless of the number of antennas. The performance gap between the conventional and proposed methods at a given $M \times N$ MIMO system with $\alpha$ of 20 is almost the same as the one with $\alpha$ of 30. Henceforth, the maximum number of iterations inside the decoder was limited to 20 for the remaining simulations. This is because we are mainly targeting to see the performance behviour of the proposed detection schemes, not that of the decoder.

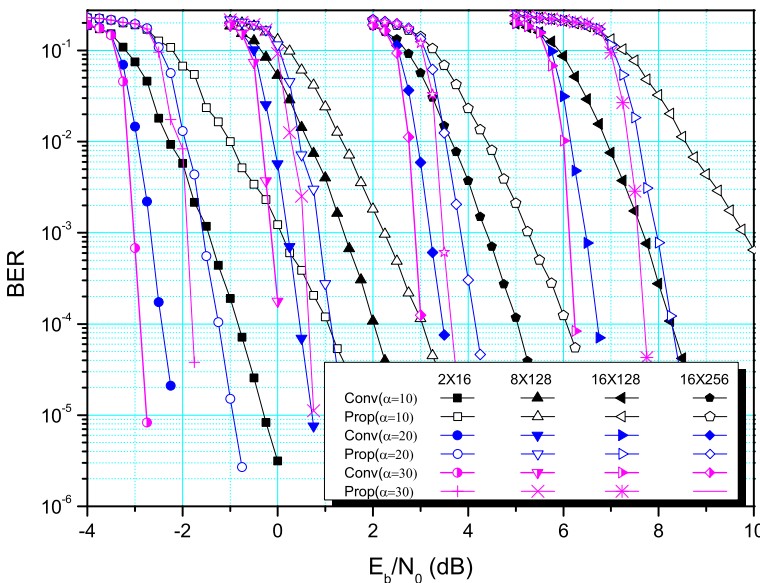

**Figure 2.** BER performance comparison according to $\alpha$ at the first iteration.

Figure 3 compares the BER performance of the $16 \times 128$ MIMO system for different $(\eta, l)$, where $\eta$ and $l$ denote the number of joint iterations and the number of detector iterations, respectively. From the simulation results in Figure 3, we can find that BER performance of the proposed method approximates to that of the conventional method as $\eta$ increases when $l = 1$. In addition, two iterations inside the detector, i.e., $l = 2$, are shown to be sufficient to produce approximating performance to that of the conventional method when $\eta = 4$. Hereafter, the maximum number of joint iterations and detector iterations were limited to 4 and 2, respectively.

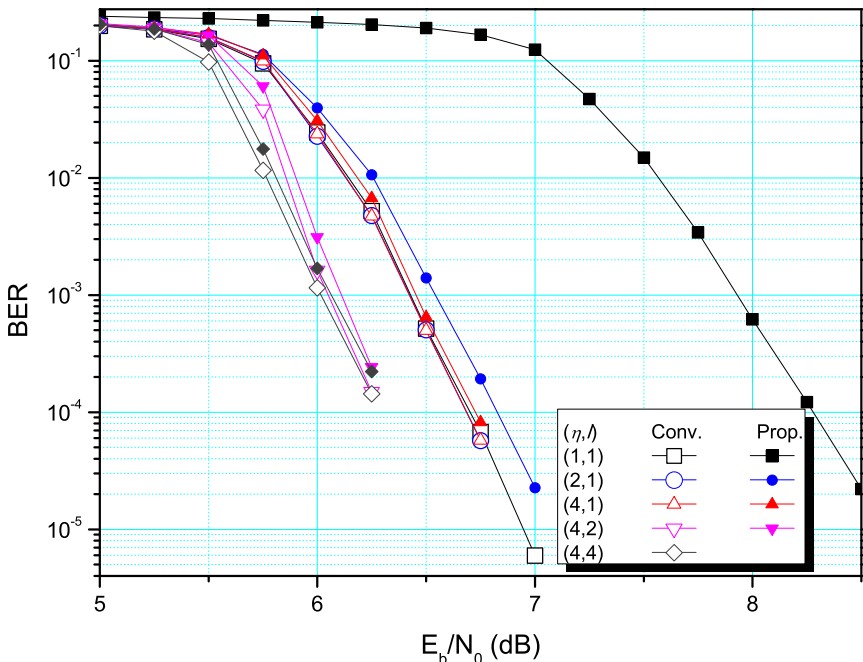

**Figure 3.** BER performance comparison with various $(\eta, l)$ for $16 \times 128$ system.

Figure 4 compares the BER performance between the proposed and conventional methods with various numbers of antennas. As shown in the figure, the performance of the proposed method slightly degrades when the number of antennas is comparatively small. On the other hand, the proposed method achieves an approximating performance to the conventional methods as the number of antennas increases. We note that we can achieve greater reduction in computational complexity with a larger number of antennas, as shown in Table 2.

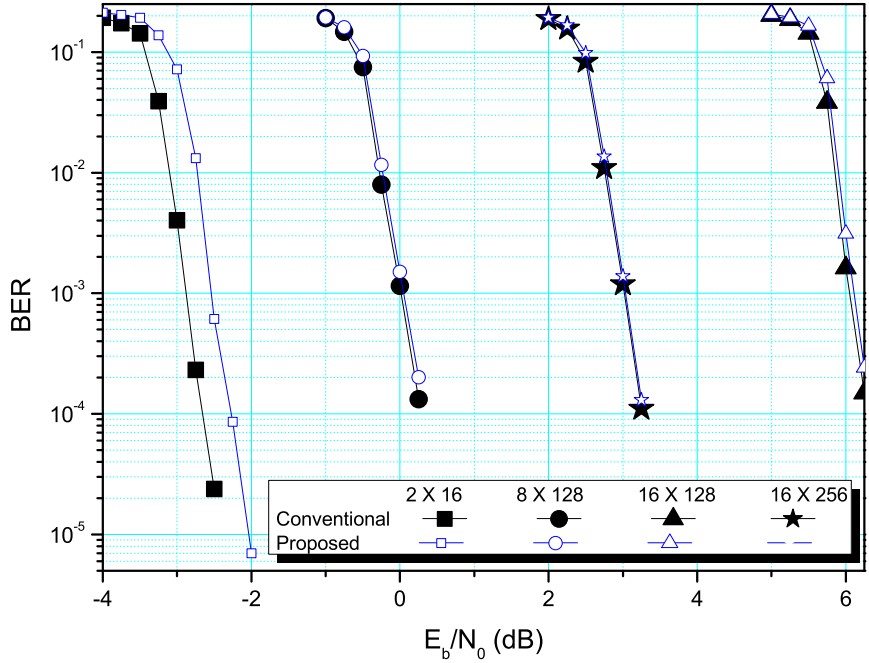

**Figure 4.** BER performance comparisons with various numbers of antennas when $(\eta, l) = (4, 2)$.

## 5. Conclusions

In this paper, we proposed an efficient method to reduce the computational complexity of the MMSE-PIC-based JIDD scheme for a massive MIMO system. The proposed method approximates the Gram matrix with a diagonal matrix composed of a universal value, and tailored it for efficient soft MIMO detection. The proposed method does not require a complex matrix inversion process and layer dependent PE-SINR estimation at every iteration, resulting in huge amount of complexity reduction. In addition, we derived compact equations to estimate soft-symbol values for the MMSE-PIC process. It was shown that the proposed method requires two orders less complexity than the conventional one. The simulation results demonstrated that the proposed method can achieve approximating performance to the conventional methods with a greatly reduced complexity.

**Author Contributions:** Conceptualization, M.Z.; Data curation, Z.Z. and S.C.; Investigation, M.Z. and S.C.; Project administration, S.K.; Software, Z.Z. and S.C.; Writing-review & editing, S.K. All authors have read and agreed to the published version of the manuscript.

**Funding:** This research was supported by the Basic Science Research Program through the National Research Foundation of Korea (NRF) funded by the Ministry of Education (2017R1D1A1B03027939) and National Natural Science Foundation of China (Grant No. 61601403).

**Conflicts of Interest:** The authors declare no conflict of interest.

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
