# Peer review of "Computationally Efficient Soft Detection Schemes for Coded Massive MIMO Systems†"

_electronics, doi:10.3390/electronics9020344_

Round 1

Reviewer 1 Report

1) Avoid using long sentences (Abstract, Introduction, Simulation results..)

2) Put more details about signals for each step in figure 1.

3) The fading channel that you used, is it multipath one or signle path?

4) Line 94: the participle 'are' is written twice.

5) Review the date of reference 5.

Author Response

We would like to thank all the reviewers for their thorough review and valuable suggestions.

In the attached file, we answered all the comments from reviewers.

We sincerely hope that this revised version and our replies meets your standards and leads to the publication of this manuscript in this journal.

Reviewer 2 Report

Reviewer’s Recommendation:

I am very sorry but this candidate paper is not accepted by my part in the present form.

Summary

The candidate paper proposes a scheme relevant to simplifying the PE-SINR estimation along with an efficient estimation method (symbol based) for reducing the computations

General comments

From a first aspect, the author seem to have worked a lot towards the presentation of their work. Unfortunately, at a second glance, this paper does not seem appropriate for a journal of electronics because its innovation (as the authors claim) is based on mathematical derivations and computations. The massive antennas systems are not presented at all and they just seem to be an application of their mathematical aspect. Furthermore, unfortunately a lot of mistakes are found in mathematical presentations and in simulated figures. Also, the simulation methods are not discussed and consequently their validity are not clear.

Suggested Improvements

Besides the aforementioned, some suggestions follow for future consideration:

Please revise abstract's expression and please write it in a simpler manner. E.g. What is the existing factor and its explanation in brief, what do you propose andd what are the results in terms of real time performance. 2 Section should be rewritten. E.g. You just mention equations from bibliography without fully justifying them. Section 2.3 is really needed? As you mention, it is already known. Why G can be approximated by N/MIm ? Where is your innovation?; Because as I understand, the previous info is found in bibliography. Equation 15 is not clearly justified. Please revise. You mention in simulations' section that you simulate your system using LDPC and code rate 1/3 and why not with Turbo Codes that they have smaller packages and and they are better in larger output rates such as 1/3 and 1/4.There is also a problem as I see here. When an approximation is done over an already approximated technique then BER with just the presence of Rayleigh fading is not enough. You should check the system in the presence of ADC and DAC whereas this means that with the injection of e.g. jitter noise your system could suffer from worst degradation compared to other already known systems. As I understand you have checked your idea in a pure theoretical manner and not theoretically with all the needed degradations in performance level. There are mistakes in Figures. E.g.: Figure explanations are not correct. The “prop” in any case is worst from the conventional one ore else your point of view is not shown properly. Please revise. Conclusions luck to show the innovation and whether a team should use your scheme and what would be the real gain. References are not adequate. Significant bibliography is not taken into consideration.

Manuscript Rating:

The paper should be rejected.

Author Response

(The authors gave the same response as above.)

Round 2

Reviewer 2 Report

The authors responded politely whereas this is very important for showing to the reviewer that they really understand his/hers position. Nevertheless, considerably changes were made but still I cannot concur towards accepting this paper. This is due that the innovation is based only in theoretical level without incorporating ADC and DACs. This is very important, as ADC and DACs in this occasion are needed for concluding the theoretical work and not as the authors declare, the implementation. The latter is something different and it involves real-time conditions. Furthermore, the mathematical innovation is not really an innovation but a derivation from previous studies. This work should be definitely appropriate even in the present form for a well-established conference but still not for a high impact journal.

Relevant to my review, if editor feels that this work should be accepted there is no problem from me but I still feel that this work is not proper for this kind of journal.

Author Response

Please refer to the attached reply letter.
